# Biological Significance of Dual Mutations A494D and E495K of the Genotype III Newcastle Disease Virus Hemagglutinin-Neuraminidase In Vitro and In Vivo

**DOI:** 10.3390/v14112338

**Published:** 2022-10-25

**Authors:** Xiaolong Lu, Tiansong Zhan, Kaituo Liu, Yu Chen, Zenglei Hu, Jiao Hu, Min Gu, Shunlin Hu, Xiaoquan Wang, Xiaowen Liu, Xiufan Liu

**Affiliations:** 1Animal Infectious Disease Laboratory, College of Veterinary Medicine, Yangzhou University, Yangzhou 225000, China; 2Jiangsu Co-Innovation Center for Prevention and Control of Important Animal Infectious Diseases and Zoonosis, Yangzhou University, Yangzhou 225000, China; 3Jiangsu Key Laboratory of Zoonosis, Yangzhou University, Yangzhou 225000, China

**Keywords:** Newcastle disease virus, vaccine strain, genotype III, hemagglutinin-neuraminidase protein, biological significance

## Abstract

As a multifunctional protein, the hemagglutinin-neuraminidase (HN) protein of Newcastle disease virus (NDV) is involved in various biological functions. A velogenic genotype III NDV JS/7/05/Ch evolving from the mesogenic vaccine strain Mukteswar showed major amino acid (aa) mutations in the HN protein. However, the precise biological significance of the mutant HN protein remains unclear. This study sought to investigate the effects of the mutant HN protein on biological activities in vitro and in vivo. The mutant HN protein (JS/7/05/Ch-type HN) significantly enhanced the hemadsorption (HAd) and fusion promotion activities but impaired the neuraminidase (NA) activity compared with the original HN protein (Mukteswar-type HN). Notably, A494D and E495K in HN exhibited a synergistic role in regulating biological activities. Moreover, the mutant HN protein, especially A494D and E495K in HN, enhanced the F protein cleavage level, which can contribute to the activation of the F protein. In vitro infection assays further showed that NDVs bearing A494D and E495K in HN markedly impaired the cell viability. Simultaneously, A494D and E495K in HN enhanced virus replication levels at the early stage of infection but weakened later in infection, which might be associated with the attenuated NA activity and cell viability. Furthermore, the animal experiments showed that A494D and E495K in HN enhanced case fatality rates, virus shedding, virus circulation, and histopathological damages in NDV-infected chickens. Overall, these findings highlight the importance of crucial aa mutations in HN in regulating biological activities of NDV and expand the understanding of the enhanced pathogenicity of the genotype III NDV.

## 1. Introduction

Newcastle disease (ND), an economically devastating infectious disease in birds worldwide, is caused by virulent Newcastle disease virus (NDV) [1]. As a single-stranded negative-sense RNA virus, NDV consists of six structural proteins and two nonstructural proteins [2]. NDV consists of two subdivisions (class I and class II) including more than twenty genotypes, although NDV has only one serotype. The class I subdivision can only be divided into 3 sub-genotypes (1.1.1, 1.1.2, and 1.2) in a single genotype 1, whereas the class II subdivision consists of 21 genotypes (genotype I–XXI) [3]. Generally, NDV virulence can be determined by the fusion protein (F) cleavage site sequence [4]. However, increasing evidence shows that the hemagglutinin-neuraminidase (HN) protein also exhibited significant effects on NDV virulence and pathogenicity [5,6].

A successful NDV infection contains the following steps: receptor recognition, cell binding, and membrane fusion [7]. As an important immunogenicity glycoprotein, the HN protein can influence these infectious steps by regulating biological functions. NDV HN proteins from different origins exhibit distinct biological activities, thereby modulating viral replication and pathogenicity [8]. The HN protein is usually manifested as a tetramer involved in receptor recognition activity [9]. The HN protein comprises cytoplasmic tail, transmembrane, stalk, and globular head regions [10]. Among these functional regions, the globular head region is involved in receptor-binding and NA activities [11,12]. The stalk region of HN is composed of four parallel helix bundles that can regulate several biological functions [13,14]. Notably, the stalk region contributes to promoting the fusion activity [15,16]. Additionally, the HN cytoplasmic tail domain is also critical for cell fusion and virion incorporation [17]. Therefore, the HN protein is a crucial molecule in biological functions, which may be involved in regulating the NDV virulence and pathogenicity.

To effectively control the ND epidemic, the prophylactic vaccination is extensively applied in poultry [18]. As an RNA virus, genetic mutations increasing virulence have been reported from time to time due to the lack of proofreading activity and the high immune pressure. A virulent NDV strain JS/7/05/Ch was highly homologous to the genotype III mesogenic vaccine strain Mukteswar [19,20]. Furthermore, Song et al. [21] identified the mutant HN protein from JS/7/05/Ch as the crucial factor for NDV pathogenicity and virulence following intravenous infection, which is consistent with results of the prior work [22]. Here, we systematically studied the effects of the mutant HN protein on biological functions of NDV and further identified A494D and E495K as the crucial factors for biological activities in vitro and in vivo. This study provides a theoretical reference for the virulence enhancement of NDV.

## 2. Materials and Methods

### 2.1. Viruses, Vectors, and Cells

Three model NDVs were provided by Prof. Xiufan Liu (Yangzhou University, China), including two parental viruses (rMukteswar and rJS/7/05/Ch) and one recombinant virus (rMukHN494 + 495^JS^). rMukteswar and rJS/7/05/Ch were generated from parental Mukteswar and JS/7/05/Ch via reverse genetics, respectively. The 494 and 495 aa in Mukteswar-type HN were mutated into those in JS/7/05/Ch-type HN, thereby generating a mutant virus rMukHN494 + 495^JS^ (Figure 1A). These recombinant viruses were stored at –70 °C after amplification. Chicken embryo fibroblasts (CEF) cells and Vero cells were preserved and cultured in our laboratory. The blunt-zero vector was purchased from TransGen Biotech, China. Besides, pCAGGS and pcDNA3.1 vectors were purchased from Miaolingbio, China.

### 2.2. Construction of the Protein Expression Plasmids and Homology Models

Mukteswar-type and JS/7/05/Ch-type HN genes were firstly amplified by a pair of primers, HN1 and HN2 (Table 1). The amplified HN fragments were successively cloned into the blunt-zero vector and pCAGGS vector with two restriction sites, *Sac*I and *Xho*I. The single-site or dual-site HN mutant was generated by substituting either or both of the nucleotide sites 1481 and 1483 of the Mukteswar-type HN gene into that of the JS/7/05/Ch-type HN gene using the fast mutagenesis kit (TransGen Biotech, China). These site-mutation HN fragments were subsequently cloned into the pCAGGS vector following the above method. These constructed plasmids bearing different HN genes were named as pCA-MukHN, pCA-JSHN, pCA-A494D, pCA-E495K, and pCA-A494D + E495K, respectively (Figure 1B). Additionally, the F gene was amplified by a pair of primers, F1 and F2 (Table 1), and was successively cloned into the blunt-zero vector and the pcDNA3.1 vector with two restriction sites, *Xho*I and *Eco*RI. The obtained F protein expression plasmid was named as pcDNA-F. These recombinant viruses and constructed clones were finally identified by Sanger sequencing (Tsingke Biotechnology, Beijing, China). The influence of mutations in HN on protein stability was predicted by the MAESTRO software. The alterations of stability predictions are based on the PDB file (PDB ID: 1e8v.1.A).

### 2.3. Hemadsorption (HAd) Assay

The adsorption activity of viruses to chicken red blood cells (CRBCs) was used to indicate the HAd activity (26). Three viruses were inoculated into Vero cells at a multiple of infection (MOI) of 0.1. The cells were washed with phosphate buffered saline (PBS) at 24 h post-infection (hpi) and overlaid with 2% CRBCs. Subsequently, cells were incubated, lysed by 0.05 M ammonium chloride, and then transferred into a new plate and measured at 549 nm. Additionally, 1 ug of plasmids expressing the HN protein were severally transfected into Vero cells. The HAd activity at the protein level was detected at 24 h post-transfection (hpt) as described above. The HAd index values of viruses or proteins were normalized to the expression of the value for rMukteswar or pCA-MukHN, which was considered 100%.

### 2.4. Neuraminidase (NA) Assay

The measurement of NA activity was performed with the neuraminidase assay kit (Beyotime Biotech, China) following the product manual. Briefly, Vero cells were harvested by centrifugation at 24 hpi following NDV infection with 0.1 MOI. The harvested cells were resuspended in the assay buffer. Subsequently, the samples were mixed with NA fluorogenic substrate, incubated at 37 °C for 30 min, and determined by an ELISA reader at 450 nm (emission wavelength) and 322 nm (excitation wavelength). Additionally, 1 ug of plasmids expressing the HN protein were severally transfected into Vero cells, followed by the determination of NA activity at the protein level at 24 hpt, as described above. The NA index values of viruses or proteins were normalized to the expression of the value for rMukteswar or pCA-MukHN, which was considered 100%.

### 2.5. Fusion Promotion Activity Assay

The fusion promotion abilities of different HN proteins were examined as described previously [23]. Briefly, three viruses were infected into Vero cells at 0.1 MOI, and then the cells were fixed in 4% paraformaldehyde after three PBS washes at 24 hpi. The syncytia formation was finally observed after staining with Giemsa solution. The fusion index was calculated as the ratio of the number of nuclei in the syncytia over the total number of nuclei in the field. Additionally, 1 ug of plasmids expressing HN proteins were individually co-transfected with pcDNA3.1-F into Vero cells, and the fusion promotion activity at the protein level was determined at 24 hpt as described above. The fusion index values of viruses or proteins were normalized to the expression of the value for rMukteswar or pCA-MukHN, which was considered 100%. 

### 2.6. Hemolytic Assessment

The viral supernatants with the identical hemagglutination unit/mL (HAU/mL) were incubated with 1% CRBC suspension for 30 min. After incubation, the mixture was harvested by centrifugation, washed, and subsequently resuspended with PBS. The samples were incubated at 37 °C for 1 h and centrifuged at 200× *g* for 5 min. Finally, the supernatants were harvested and determined at 549 nm by an ELISA reader. NH_4_OH (0.03 M) and the PBS treatment group were considered the positive and negative control, respectively. The hemolytic index values were each presented as the percentage of the value for rMukteswar at 2^6^ HAU/mL (considered as 100%).

### 2.7. Western Blot

Firstly, three viruses were inoculated into CEF cells at 0.1 and 1 MOI. Afterwards, the cells were lysed in RIPA buffer with the proteinase inhibitor PMSF (Beyotime Biotech, China) at 24 hpi. Total protein concentration of the harvested cells was then detected by the BCA assay. The denatured protein was detected using 10% SDS-PAGE and further transferred to polyvinylidene difluoride (PVDF) membranes. Subsequently, the PVDF membranes were blocked and incubated with the diluted primary and secondary antibodies. The antibodies used in this research were as follows: HN monoclonal antibody (Santa Cruz Biotechnology, USA), F monoclonal antibody, NP monoclonal antibody (provided by our lab), the indicated protein (Sigma-Aldrich, USA), and anti-mouse IgG antibody (TransGen Biotech, China). The detection was performed by incubating the membrane with chemiluminescent substrate and exposing the membrane in a dark room by using ChemiDoc Imagers (Bio-Rad Laboratories, USA). Finally, gray bands were evaluated by ImageJ 1.48v software.

### 2.8. Viral Growth Kinetics

Three viruses were inoculated into CEF cells at 0.1 and 1 MOI. The unattached viruses were removed after adsorption. The cells were then washed and cultured for the indicated time points. The cell supernatant was finally gathered and determined by the 50% tissue culture infective dose (TCID_50_). 

### 2.9. Cell Viability

CEF cells cultured in 96-well plates were infected with three viruses at 0.1 and 1 MOI for the indicated time points. The infected cells were incubated with CCK-8 solution (CCK-8 assay kit, Beyotime Biotech, China). Absorbance was finally measured at 450 nm after incubation for 1 h.

### 2.10. Immunofluorescence Assay (IFA)

Vero cells were transfected with 1 μg of each plasmid expressing the HN protein in 12-well plates. The transfected cells were washed, fixed, and blocked, successively. Then, the cells were inoculated with the anti-NDV HN primary antibody at 4 °C for 12 h and secondary antibody-conjugated FITC (TransGen Biotech, China) at 37 °C for 1 h. The cells were subsequently analyzed and photographed by a fluorescence microscope. Fluorescence intensity was evaluated by ImageJ 1.4 software.

### 2.11. Animal Experiments

We further carried out the in vivo challenge test to investigate the biological significance of dual aa mutations in vivo. Three viruses were inoculated into nine four-week-old chickens with 10^6^ 50% egg infectious dose (EID_50_) of virus per chicken through the intravenous route, respectively. Six chickens received an equal amount of PBS as a negative control. The health status of each chicken was observed and recorded daily for 10 days. Peripheral blood was collected from the wing vein of chickens at 1, 3, and 5 dpi. The RNA of peripheral blood was thereafter extracted and reverse transcribed into cDNA. The mRNA levels of the NDV NP gene in peripheral blood were determined by qPCR. Three chickens were euthanized at 4 dpi in each group, and the lung, spleen, and thymus from these chickens were gathered to evaluate the histopathological changes. Additionally, oropharyngeal and cloacal swabs were gathered daily from the remaining chickens to evaluate virus shedding. The collected swabs were scrubbed and centrifuged, followed by inoculating monolayers of CEF cells. Then, cell supernatants were collected at 96 hpi and determined by the hemagglutination (HA) assay.

### 2.12. Histopathology

The lung, spleen, and thymus samples were collected as described above. These samples were fixed with neutral formalin solution, followed by being dehydrated, paraffin-embedded, sectioned, baked, dewaxed, and stained with hematoxylin and eosin. After the preparation of pathological sections, histopathological changes were observed and photographed under a light microscope.

### 2.13. Statistical Analyses

The one-way or two-way analysis of variance (ANOVA) was used to determine the statistical significance. The statistics were evaluated with GraphPad Prism 7.00 software. *p* < 0.05 was considered statistically significant.

## 3. Results

### 3.1. Acquisition of the Mutants and Simulation of the Amino Acid Autations

The whole genome sequences of three rNDVs were verified using PCR and Sanger sequencing. The sequencing results confirmed the differential nucleotide and amino acid (aa) sequences between rNDVs: fourteen nucleotide and seven aa differences between rMukteswar and rJS/7/05/Ch, two nucleotide and two aa differences between rMukteswar and rMukHN494 + 495^JS^, and twelve nucleotide and five aa differences between rJS/7/05/Ch and rMukHN494 + 495^JS^. Five eukaryotic expression plasmids for HN mutants were successfully generated and characterized by Sanger sequencing (Figure 1C). To evaluate the stability and de-stability of HN mutants, four aa mutations, M145T, V266I, A494D, and E495K, were evaluated by the MAESTRO software. As shown in Table 2, these mutations were ranked by predicted changes in free energy (ΔΔG values). The predicted results showed that V266I and E495K were identified as the stabilizing mutations, while M145T and A494D were identified as the destabilizing mutations. Notably, both A494D and E495K resulted in higher ΔΔG values, indicating a stronger influence on the HN protein stability. Effects of these mutations on the HN protein structure may be involved in regulating the protein function.

### 3.2. Analysis of Biological Activities of the Mutant HN Proteins at the Protein Level

To investigate differences in biological functions of HN proteins, we used a series of HN expression plasmids (pCA-MukHN, pCA-JSHN, pCA-A494D, pCA-E495K, and pCA-A494D + E495K) to determine their HAd (Figure 2A), NA (Figure 2B), and fusion promotion activities (Figure 2C) following the transfection of HN proteins. pCA-JSHN exhibited increased HAd and fusion promotion activities, but decreased NA activity compared with pCA-MukHN. Briefly, pCA-JSHN showed a 34% increase in the HAd activity (*p* < 0.001), a 68% increase in the fusion promotion activity (*p* < 0.0001), and a 24% decrease in the NA activity (*p* < 0.001) compared with pCA-MukHN. Furthermore, we detected the roles of two aa substitutions, A494D and E495K, located in the antigenic epitope in biological activities. Neither pCA-A494D nor pCA-E495K exhibited a significant difference in HAd, NA, and fusion promotion activities compared with pCA-MukHN. However, pCA-A494D + E495K showed a significant increase in HAd and fusion promotion activities, and a significant decrease in the NA activity compared pCA-MukHN, which was close to pCA-JSHN. Therefore, pCA-A494D + E495K showed a stronger regulation of biological activities than pCA-A494D and pCA-E495K, indicating the synergistic effects of A494D and E495K on biological activities. In detail, pCA-A494D + E495K showed a 28% increase in the HAd activity (*p* < 0.001), a 52% increase in the fusion promotion activity (*p* < 0.001), and a 22% decrease in the NA activity (*p* < 0.01) compared with pCA-MukHN. Therefore, the mutant HN protein is involved in regulating biological activities, and A494D and E495K play a synergistic role.

### 3.3. Expression Efficiency of Different HN Proteins

The IFA assay was used to evaluate the expression efficiency of HN proteins. The original and mutant HN proteins were all efficiently expressed in Vero cells. The quantitative results showed no significant difference in the expression efficiency of different HN proteins (Figure 3). Thus, the distinct biological activities among various HN proteins are caused by differences in expression efficiency.

### 3.4. Assessment of HAd and NA Activities of NDVs Bearing Different HN Proteins

The receptor-binding and NA activities of viruses bearing the original and mutant HN proteins were measured by HAd and NA assays. As illustrated in Figure 4, rJS/7/05/Ch induced an 87% increase in the HAd activity (*p* < 0.0001) and a 58% decrease in the NA activity (*p* < 0.001) compared with rMukteswar. Noteworthy, rMukHN494 + 495^JS^ also mediated a higher HAd activity (a 61% increase, *p* < 0.001) but a lower NA activity (a 48% decrease, *p* < 0.01) than rMukteswar, which was similar to rJS/7/05/Ch. Hence, A494D and E495K in HN are crucial for the receptor-binding and NA activities at the virus level.

### 3.5. Evaluation of Fusogenic and Hemolytic Activities of NDVs Bearing Different HN Proteins

The membrane fusion activity of viruses bearing the original and mutant HN proteins was detected for the fusion indices and hemolytic activities. As illustrated in Figure 5A, the parental and recombinant viruses caused the severe syncytia compared with the uninfected group. The quantification of the fusion activity of the three viruses showed that both rJS/7/05/Ch and rMukHN494 + 495^JS^ significantly strengthened the fusion activity compared with rMukteswar (*p* < 0.0001 and 0.001, respectively). Specifically, rJS/7/05/Ch (199%) and rMukHN494 + 495^JS^ (182%) formed more syncytia than rMukteswar (considered as 100%). As shown in Figure 5B, rJS/7/05/Ch and rMukHN494 + 495^JS^ significantly strengthened the hemolytic activity compared with rMukteswar, ranging from 2^2^ to 2^6^ HAU/mL. Besides, rJS/7/05/Ch and rMukHN494 + 495^JS^ showed similar fusion indices and hemolytic activities. Therefore, A494D and E495K in HN have pivotal effects on the fusogenic and hemolysis activities at the virus level.

### 3.6. Evaluation of the F Protein Cleavage Activity of NDVs Bearing Different HN Proteins

Effects of the mutant HN protein on the F protein cleavage activity were detected by Western blotting. As illustrated in Figure 6, F proteins of the three viruses were all cleaved into F_1_ after infection at an MOI of 0.1 and 1. Notably, the rJS/7/05/Ch and rMukHN494 + 495^JS^ groups showed higher F_1_/HN ratios than the rMukteswar group, significantly at 1 MOI, indicating a stronger activation of the F protein. Besides, the F_1_/HN ratios between NDVs bearing A494D and E495K showed no obvious difference at each MOI. These results suggest that A494D and E495K in HN promote the F protein cleavage, thereby enhancing the activation of the F protein.

### 3.7. Determination of Cell Viability and Virus Replication In Vitro following Infection with NDVs Bearing Different HN Proteins

We next determined the effects of the mutant HN proteins on cell survival and virus replication in vitro. As shown in Figure 7A, both NDVs bearing A494D and E495K infection decreased the cell viability compared with rMukteswar at each MOI. Briefly, rJS/7/05/Ch and rMukHN494 + 495^JS^ significantly reduced the cell viability from 12 hpi at 0.1 MOI and from 6 hpi at 1 MOI, respectively. The virus growth curves indicated that rJS/7/05/Ch and rMukHN494 + 495^JS^ mediated a stronger replication ability than rMukteswar at the early stage of infection in CEF cells, while rMukteswar replicated more efficiently than the other two viruses after 24 or 36 hpi (Figure 7B). The differences in virus replication were in a dose-dependent manner. Additionally, those NDVs bearing A494D and E495K mediated similar cell survival and virus replication levels. 

These findings indicate that A494D and E495K pose important effects on cell viability and virus replication in vitro.

### 3.8. Pathogenicity of NDVs Bearing Different HN Proteins in Four-Week-Old Chickens

To validate the functions of A494D and E495K in HN on viral pathogenicity, the pathogenicity, virus shedding, and virus replication in NDV-infected chickens were evaluated. The animal experiments showed that the clinical symptoms and mortality rates of the three infected groups were quite variable. Briefly, both rJS/7/05/Ch- and rMukHN494 + 495^JS^-infected chickens showed obvious and quick clinical signs, such as breathing difficulty, lethargy, and decreased appetite, while the rMukteswar-infected chickens only exhibited mild symptoms such as sneezing and coughing until the late stage of infection. In the rJS/7/05/Ch- and rMukHN494 + 495^JS^-infected groups, the morbidity and mortality rates were up to 100% (6/6), and all chickens died by 6 and 8 dpi, respectively. However, the morbidity was only 17% (1/6) in the rMukteswar-infected group, and none of the infected chickens in the rMukteswar-infected group died during the observation period (Figure 8A,B).

To compare the virus-shedding abilities of three viruses in chickens, oropharyngeal and cloacal swabs were collected for viral determination for 10 days. As shown in Table 3, both NDVs bearing A494D and E495K mediated stronger virus-shedding abilities in chickens than rMukteswar. Briefly, the oropharyngeal and cloacal swabs in rJS/7/05/Ch- and rMukHN494 + 495^JS^-infected chickens were positive from 2 and 3 dpi, respectively. Moreover, all the chickens exhibited positive oropharyngeal and cloacal swabs in the rJS/7/05/Ch and rMukHN494 + 495^JS^ groups until death. In contrast, virus shedding in the rMukteswar group was later than the other two groups. Simultaneously, virus shedding was attenuated rapidly during the rMukteswar infection, and no virus shedding was detected at the late stage of rMukteswar infection. Mukteswar and JS/7/05/Ch showed significant differences in virulence after intravenous infection, and thus we further determined viral replication levels of the three viruses in peripheral blood of chickens. As shown in Figure 8C, both NDVs bearing A494D and E495K induced higher NP mRNA levels in peripheral blood of chickens than rMukteswar during the infection. This interesting result also indicated that dual mutations A494D and E495K helped the mutant virus to be maintained in peripheral blood longer than the original virus. Additionally, the PBS-inoculated chickens remained NDV-negative throughout the experiment. Therefore, A494D and E495K contribute to enhancing the pathogenicity, virus shedding, and circulation in vivo.

### 3.9. Histopathological Changes in the Lung, Spleen, and Thymus following NDV Infection

Compared with the mock group, three NDV-infected groups exhibited histopathological damages. As for infection with rJS/7/05/Ch and rMukHN494 + 495^JS^, bleeding, severe abnormality, massive alveoli atrophied, pulmonary parenchyma, and massive inflammatory cell infiltration were observed in the infected lungs. More significant pathological damages in the infected spleens included massive inflammatory cell infiltration, necrosis, lymphocyte damage, tissue fibrosis, and red blood cell exudation. Simultaneously, the main histopathological manifestations were high-intensity lymphodepletion in the thymus. However, only slight histopathological damages were observed in the rMukteswar-infected group, such as slight structural changes and bleeding (Figure 9).

## 4. Discussion

The control of ND includes strict biosecurity to prevent the introduction of virulent NDV onto poultry farms and proper administration of vaccines. The ND vaccine is widely administered in poultry in several countries around the world, including China [24]. Mukteswar, the classical I ND vaccine, was once widely used as an emergency immunization vaccine. However, RNA viruses are prone to mutation and thereby change viral virulence due to the lack of proofreading activity and the high immune pressure [25]. Prior research has reported that Mukteswar evolves into a virulent NDV strain JS/7/05/Ch during vaccination of poultry [19]. The vaccine variant strain JS/7/05/Ch shows high genomic similarity with Mukteswar, and their major amino acid mutations concentrate on HN [20]. HN is well-known as a multifunctional protein involved in various biological activities [26].

Protein structural stability poses an important role in protein functionality [27,28]. The aa mutations in HN, especially A494D and E495K, caused the alteration of the protein structural stability, which might largely affect the function of HN. We subsequently investigated the effects of the mutant HN protein on biological functions of NDV. Generally, virus–receptor recognition is required for viral infection [29]. As for NDV, the receptor-binding activities can be determined by the HAd assay. The HN protein is able to recognize and bind to sialic acid receptors on the RBC surface, and therefore NDV exhibits the ability of hemagglutinating RBCs [30]. Here, the mutant HN protein significantly enhanced the receptor-binding activity of NDV, which could be conducive to initial infection of the virus. Following the virus–receptor binding, NDV invades into host cells by the membrane fusion process. The optimum membrane fusion needs the synergistic effects of F and HN proteins [31]. This study demonstrated that the mutant HN protein enhanced the syncytium formation ability of NDV. Moreover, red cell membrane rupture is one of the most essential reasons for hemolysis to occur [32]. Here, the mutant HN protein improved the hemolysis of RBCs, also indicating the enhanced membrane fusion of RBCs. Thus, the strong membrane fusion activity may be one of the important factors leading to the enhanced infection ability of genotype III NDV. The NDV F protein is initially synthesized as an inactive precursor, F_0_, and subsequently cleaved into F_1_ and F_2_ subunits, gaining the fusion ability [33]. Multiple studies have demonstrated that the cleavage efficiency of NDV F protein can be influenced by different origins of HN proteins [8,34,35]. According to these published studies, we considered that the mutant HN protein affected the F protein cleavage in this study, consequently resulting in differential fusion activities. The further assay showed that the mutant HN protein promoted the F protein cleavage, which was beneficial to the activation of NDV F protein. Accordingly, the enhanced activation of F protein could contribute to boosting membrane fusion of JS/7/05/Ch. 

The NA activity of NDV HN protein resembles that of the NA protein of influenza viruses, which cleaves off sialic acid on the cell surface and thereby enables the release of progeny viruses from host cells [36]. This study showed that the mutant HN protein impaired the NA activity of JS/7/05/Ch, which might limit the release of progeny viruses. Compared with Mukteswar, JS/7/05/Ch exhibited a higher replication level at the early stage of infection but a lower replication level after 12 hpi. Thus, the decreased virus titer of JS/7/05/Ch after 12 hpi might be partly attributable to its weakened NA activity. Generally, viruses need to hijack the host to facilitate survival and replication, and thus cell damage and death can limit virus replication [37]. We found that JS/7/05/Ch caused more rapid cell death than Mukteswar. Generally, excessive virus replication can result in a quick death of the host, which is adverse to virus transmission [38]. Herein, the rapid viral proliferation of JS/7/05/Ch could exacerbate the cellular injury and death, thereby limiting the virus growth at the late stage of infection. However, the specific mechanism of how the HN mutant regulates cell survival remains to be further studied. To sum up, JS/7/05/Ch-type HN provided the higher titer of the virus at the early stage of infection based on the enhanced receptor-binding activity, fusion activity, and F protein activation. Whereas the impaired NA activity and cell viability could limit virus replication of JS/7/05/Ch at the late stage of infection. Accumulating studies have identified some specific amino acids responsible for biological activities [10,35,39]. For instance, four aa mutations (G54S, F110L, G116R, and A469V) in HN cause variation in biological activities, and the residues 315 and 369 in HN facilitate the thermostability of NDV. Furthermore, residues 494 and 495 in NDV HN are reported to be located in the antigen epitope, and thereby involved in receptor recognition [40,41,42]. Coincidentally, Mukteswar and JS/7/05/Ch showed aa differences at residues 494 and 495. Here, A494D and E495K in HN were identified to synergistically influence in vitro biological activities of NDV, which can be responsible for the differential biological activities between the genotype III NDVs. 

Multiple studies have shown that mutations in the HN protein are considered as determinants of in vivo biological activities, such as virulence and pathogenicity [43,44]. We have previously demonstrated that the aa mutations in the HN protein were required for the virulence and pathogenicity of genotype III NDV, but the research content needs to be further enriched. In this study, we further enriched the effects of A494D and E495K on NDV-induced in vivo biological activities, such as case fatality rates, virus shedding, and circulation abilities in chickens. As expected, those NDVs bearing A494D and E495K mediated the higher morbidity and mortality of chickens. Simultaneously, A494D and E495K enhanced the virus-shedding levels following NDV infection. It is reported that controlling the efflux of viruses contributes to limiting the spread and infection of viruses [45]. Therefore, the premature and lasting virus shedding can facilitate the transmissibility of genotype III NDV, thus contributing to virus infection. Notably, dual mutations A494D and E495K facilitated NDV replication in peripheral blood, indicating a longer virus maintenance in peripheral blood. Generally, the longer presence of the virus in the vector would be conducive to its infection. Hence, genotype III NDVs bearing A494D and E495K can efficiently infect the host and enhance the pathogenicity to chickens. This interesting result can help clarify the reason for the difference in the virulence of genotype III NDV after intravenous inoculation. Nevertheless, the precise reason why the mutant viruses can be maintained in peripheral blood for longer needs to be further explored in the future. In addition, those NDVs bearing A494D and E495K exhibited stronger tissue tropism and histopathological damages, especially including massive inflammatory cell infiltration and lymphocyte damages. Inflammatory cell infiltration and lymphocyte depletion would cause the collapse of the immune system and ultimately lead to the enhancement of pathogenicity [46,47]. Therefore, significant histopathological damages can be essential for the NDV pathogenicity. 

The balance among in vitro biological activities can pose an important role in viral infection and pathogenicity of NDV [8]. Here, our findings further highlighted the importance of the HN-regulated in vitro biological activities in modulating pathogenicity and infection of NDV in vivo. Moreover, we demonstrated that A494D and E495K in the HN protein exhibited a crucial factor in the association between in vitro and in vivo biological activities of the genotype III NDV.

## 5. Conclusions

In summary, this study systematically illustrated that the mutant HN protein significantly influences the biological activities of NDV in vitro and in vivo. Dual aa mutations A494D and E495K in the HN protein were identified as the crucial factors for the differential biological activities. The balance among in vitro biological activities has come forth as a critical regulator of infection of NDV in vivo. These data augment our understanding underlying the biologic mechanism of NDV and contribute to explaining the enhanced pathogenicity of genotype III NDV.

## Figures and Tables

**Figure 1 viruses-14-02338-f001:**
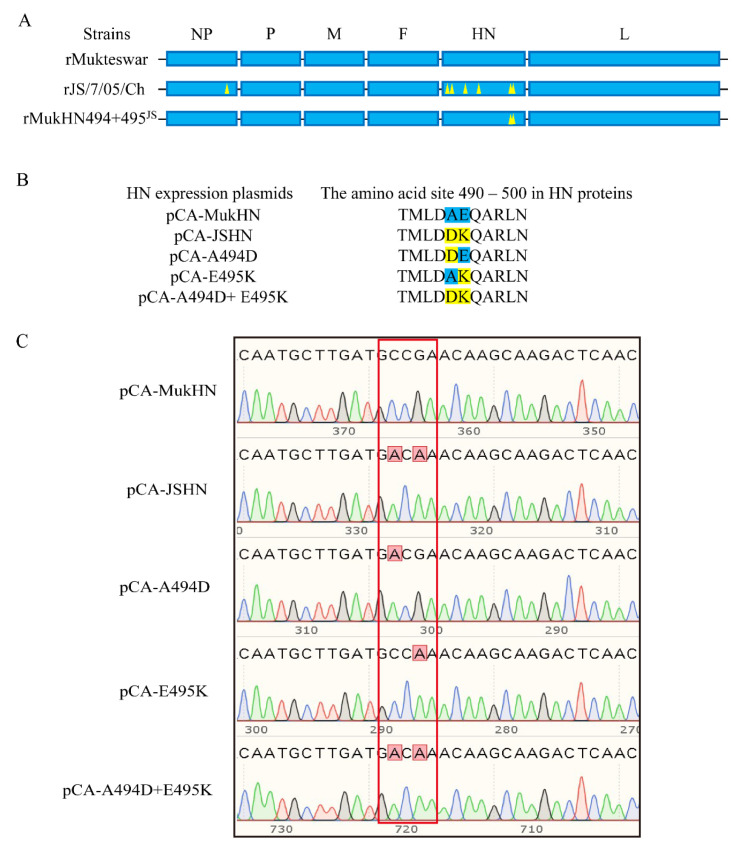
Schematic representation of the mutants used in this study. (**A**) Three NDV strains were used as model viruses in this study, including the vaccine strain rMukteswar, the vaccine variant strain rJS/7/05/Ch, and the dual-site mutant strain rMukHN494 + 495^JS^. rMukHN494 + 495^JS^ was the rMukteswar derivative bearing the 494 and 495 aa of rJS/7/05/Ch. The aa mutations in the HN protein of NDV are marked with yellow triangles, including P438S in NP, and N19S, S29T, M145T, V266I, A494D, and E495K in HN. (**B**) Schematic representation of the aa mutations ranging from position 490 to 500 of different HN expression plasmids. Residues in blue and yellow indicate amino acids from Mukteswar and JS/7/05/Ch, respectively. (**C**) Constructed clones were identified by Sanger sequencing. The sequences of key mutation sites were displayed in red frames using the SnapGene 3.2.1 software.

**Figure 2 viruses-14-02338-f002:**
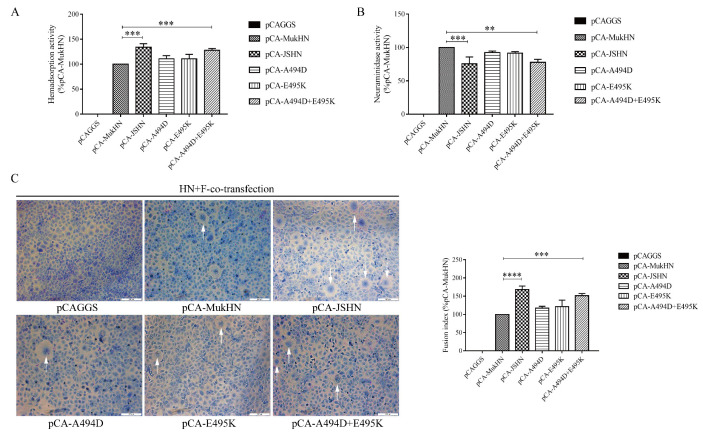
Biological activities of HN proteins at the protein level. HAd (**A**) and NA (**B**) activities were determined after transfection with each HN protein expression plasmid. (**C**) The fusion promotion activity of different HN proteins was measured following co-transfection with each HN protein expression plasmid and the F protein plasmid. Images were taken under a microscope at ×200 magnification. The syncytia formation is indicated by white arrows. Bar indicates 200 µm. All values are normalized to the expression of the values for pCA-MukHN, which was set at 100%. *** p* < 0.01, **** p* < 0.001, and ***** p* < 0.0001.

**Figure 3 viruses-14-02338-f003:**
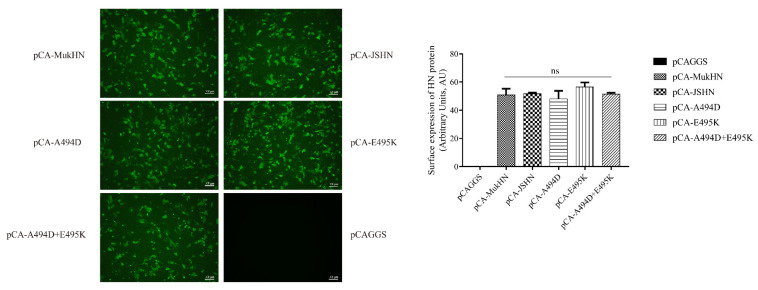
Analysis of expression efficiency of different HN proteins. Different HN protein expression plasmids were transfected into Vero cells, followed by incubating with the HN protein primary antibody and corresponding secondary antibody. Images were then taken by a fluorescent microscope at ×100 magnification. Bar indicates 100 µm. IFA results were evaluated by the ImageJ software. ns indicates no significance.

**Figure 4 viruses-14-02338-f004:**
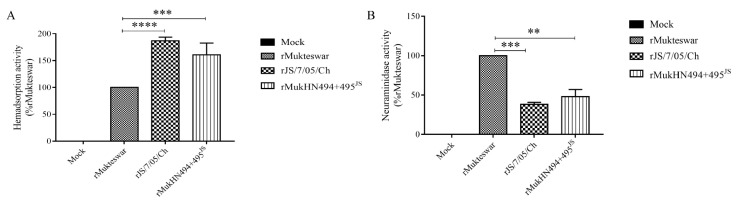
HAd and NA activities of NDVs bearing different HN proteins. Vero cells were infected with three viruses at 0.1 MOI for 24 h, and then detected for the HAd activity (**A**) and the NA activity (**B**), respectively. HAd and NA activities of NDV are normalized to the expression of the values for rMukteswar, which was set at 100%. ** *p* < 0.01, *** *p* < 0.001, and **** *p* < 0.0001.

**Figure 5 viruses-14-02338-f005:**
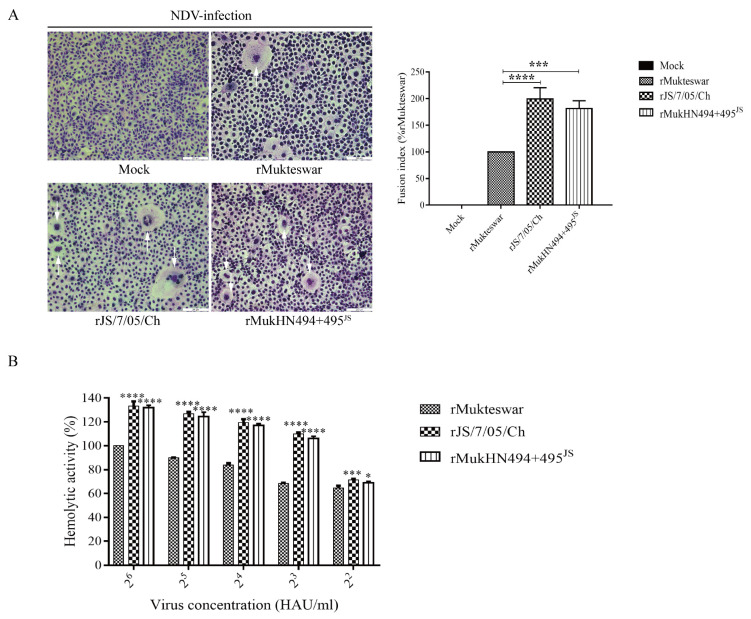
Syncytium formation and hemolytic activities of NDVs bearing different HN proteins. (**A**) Vero cells were infected with three viruses at 0.1 MOI for 24 h, and then detected for the syncytia formation. Images were taken under a microscope at ×200 magnification. White arrows were used to mark the syncytia formation. Bar indicates 200 µm. (**B**) The hemolytic activity was compared among the parental and recombinant viruses. Statistical significance was compared with that of rMukteswar. All values are normalized to the expression of the values for rMukteswar, which was set at 100%. ** p* < 0.05, **** p* < 0.001, and ***** p* < 0.0001.

**Figure 6 viruses-14-02338-f006:**
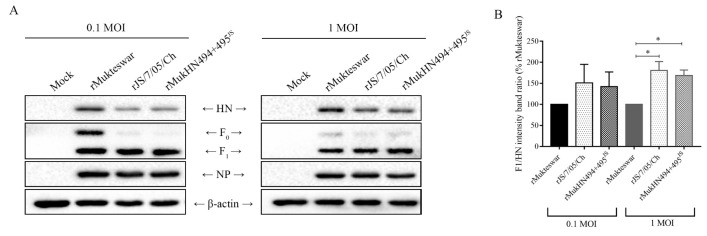
Evaluation of F protein cleavage activities of NDVs bearing different HN proteins. (**A**) Three viruses were inoculated into CEF cells at 0.1 and 1 MOI, respectively. The amounts of HN, F_0_, and F_1_ proteins were detected by Western blotting and then the cleavage-promotion activity of different HN proteins was analyzed. (**B**) The gray bands were evaluated by the ImageJ software. The F_1_/HN ratio at each infected group was normalized to that of the rMukteswar group. ** p* < 0.05.

**Figure 7 viruses-14-02338-f007:**
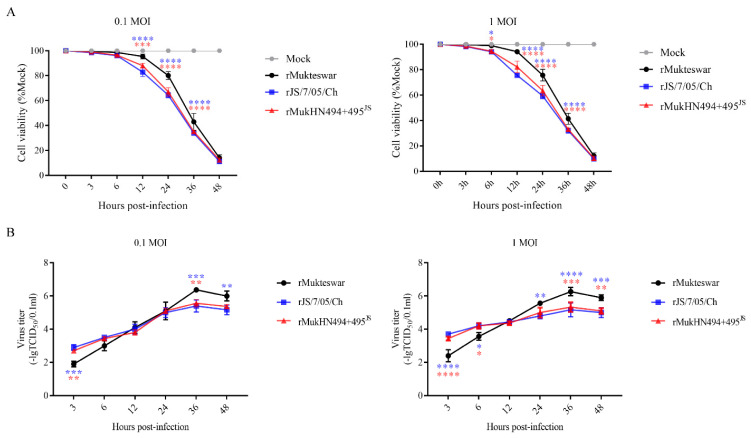
Evaluation of cell viability and virus replication ability in NDV-infected CEF cells. Three viruses were inoculated into CEF cells at 0.1 and 1 MOI, respectively. (**A**) Reduction of cell viability in NDV-infected CEF cells at 3, 6, 12, 24, 36, and 48 hpi. (**B**) Growth characteristics of the three viruses were determined during 3–48 hpi. All values were compared with that of rMukteswar. ** p* < 0.05, *** p* < 0.01, **** p* < 0.001, and ***** p* < 0.0001.

**Figure 8 viruses-14-02338-f008:**
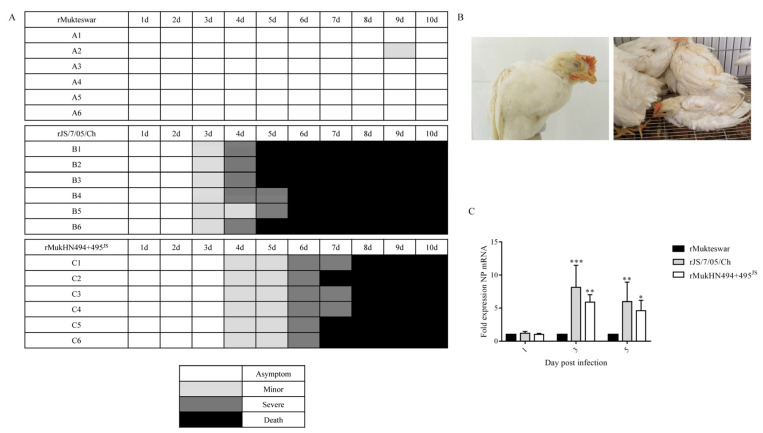
Pathogenicity of NDVs bearing different HN proteins. (**A**) Clinical signs were observed and recorded once daily. Clinical scores of chickens were calculated according to degree of severity. (**B**) Clinical symptoms of diseased chickens. rJS/7/05/Ch-infected chickens at 5 dpi (left) and rMukHN494 + 495^JS^-infected chickens at 6 dpi (right). Both rJS/7/05/Ch- and rMukHN494 + 495JS-infected chickens showed serious clinical symptoms, including breathing difficulty, lethargy, and decreased appetite. (**C**) The mRNA levels of the NDV NP gene in peripheral blood of chickens at 1, 3, and 5 dpi. All values were compared with that of rMukteswar. ** p* < 0.05, *** p* < 0.01, and **** p* < 0.001.

**Figure 9 viruses-14-02338-f009:**
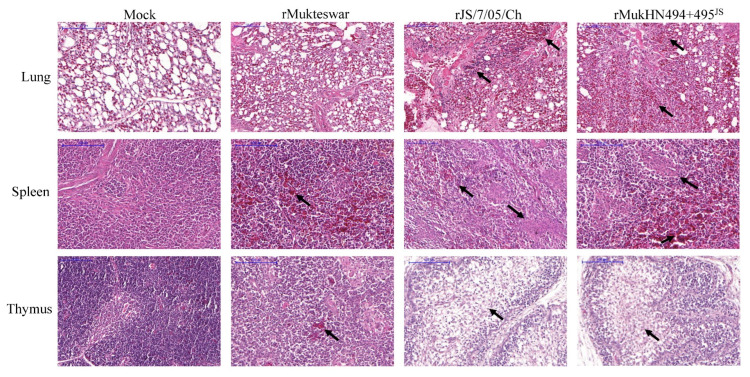
Histopathology of the lung, spleen, and thymus at 4 dpi. The obvious lesions are marked with black arrows. Magnification 200×, scale bar 100 μm.

**Table 1 viruses-14-02338-t001:** Primer sequences used to amplify the HN and F sequences of the NDV genome.

Primer	Sequence (5′–3′)
HN1	**CGAGCTCG**ATGGACAGCGCAGTTAGCCAAG (*Sac*I)
HN2	**CCTCGAGG**TTAAACCCCACCATCCTTGAG (*Xho*I)
F1	**CCTCGAGG**ATGATCTGTCTTGATTACTTACAGC (*Xho*I)
F2	**GGAATTCC**CATTTTTGTAGTAGCTCTCATCTG (*Eco*RI)

Sequences shown in bold indicate recognition sites of restriction enzymes.

**Table 2 viruses-14-02338-t002:** Prediction of stabilizing and destabilizing mutations in the HN proteins using the MAESTRO algorithm. ΔΔG_pred, predicted change in free energy (kcal/mol). Negative and positive values indicate a stabilizing and destabilizing mutation, respectively. c_pred, confidence estimation (0–1) of the prediction.

Mutant	ΔΔG_pred	c_pred
E495K	−0.12755	0.884864
V266I	−0.03536	0.86262
A494D	0.199226	0.900809
M145T	0.019491	0.838821

**Table 3 viruses-14-02338-t003:** Virus shedding in oropharyngeal and cloacal swabs of chickens, determined by inoculating monolayers of CEF cells. ^a^ oropharyngeal swabs, ^b^ cloacal swabs.

Days Post-Infection	No. of Chickens Shedding/Total No. of Chickens							
	Virus							PBS
	rMukteswar		rJS/7/05/Ch		rMukHN494 + 495^JS^			
	O ^a^	C ^b^	O	C	O	C	O	C
1	0/6	0/6	0/6	0/6	0/6	0/6	0/3	0/3
2	0/6	0/6	6/6	0/6	6/6	0/6	0/3	0/3
3	6/6	0/6	6/6	6/6	6/6	3/6	0/3	0/3
4	3/6	4/6	6/6	6/6	6/6	5/6	0/3	0/3
5	1/6	4/6	2/2	2/2	6/6	6/6	0/3	0/3
6	0/6	2/6	-	-	6/6	6/6	0/3	0/3
7	0/6	2/6	-	-	3/3	3/3	0/3	0/3
8	0/6	1/6	-	-	-	-	0/3	0/3
9	0/6	1/6	-	-	-	-	0/3	0/3
10	0/6	0/6	-	-	-	-	0/3	0/3

- means that all infected chickens are dead.

## Data Availability

Not applicable.

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
