# Peer review of "Biological Significance of Dual Mutations A494D and E495K of the Genotype III Newcastle Disease Virus Hemagglutinin-Neuraminidase In Vitro and In Vivo"

_viruses, 2022, doi:10.3390/v14112338_

Round 1
Reviewer 1 Report
Comments to authors
The authors elucidated that dual mutations at positions 494 and 495 of HN protein of genotype III Newcastle disease virus were involved in viral characters such as HAd, fusion, F cleavage, and NA activities. The virus bearing 494/495 aa mutations more replicated in tissue culture at early stage of infection than its original virus. The mutant virus also showed higher intravenous pathogenicity in chickens. Based on the results, authors concluded that these mutations are crucial factors for the biological activities of genotype III NDV viruses.
The context is simple and clear, however, following points should be revised/reconsidered.
Major comments
1. Authors should strictly discuss the results based on the significant differences by statistical analyses at sections as follows.
L320-L323, Figure 2: HN with single mutations (A494D or E495K) showed no significant differences from their control, MukHN, in HAd, NA, and fusion activities. The description in L320-L323 should be revised.
L407-L410, Figure 6: Significant differences in F1/HN ratios were ONLY found in the case of MOI 1 viral infection. The description in L407-L410 should be revised.
L434, Figure 7B: Significant differences were found after 24 or 36 hpi, not 12 hpi.
2. L485-L487, L626-L628, L637: Tissue tropisms of the viruses were hard to discuss based on the present experimental infection study because the chickens were inoculated with the virus via direct intravenous route, not natural infection route. Authors should describe not ‘circulation route’ but the reason why the mutant viruses maintained in peripheral bloods longer than the original virus.
3. The aa mutations at the 494 and/or 495 of HN proteins have been also found in other (genotype III) NDVs in the field? Such information/discussion should be added to make this article more significant.
Minor comments
Introduction: Additional explanation about NDV genotype is preferable because ‘Genotype III’ should be one of keywords of this paper.
L95, L106: ‘Kept in our lab’ and ‘provided by our lab’ are not appropriate description for academic article because readers are not able to reproduce your work. Please clarify backgrounds or previous reference works of these materials. For example, you isolated the viruses by yourself or obtained them from other institutes? Vectors were purchased/handmade/provided from others? etc.
Table 1: Recognition sites of restriction enzymes should be also shown in the table.
Table 2: E494K -> E495K?
L341: ‘*’ not be Italic
L350-L355: Font is different from other sections.
L426: ‘greatly’ should be deleted. Too subjective.
Figure 8B: should be deleted. Same results as Figure 8A.
Figure 8C: not informative pictures. If you want to show these, detail explanations about symptoms should be added in the figure legends.
Discussion: should be divided into some paragraphs according to the contents.
Author Response
Response to Reviewer 1 Comments
Comments to authors:
The authors elucidated that dual mutations at positions 494 and 495 of HN protein of genotype III Newcastle disease virus were involved in viral characters such as HAd, fusion, F cleavage, and NA activities. The virus bearing 494/495 aa mutations more replicated in tissue culture at early stage of infection than its original virus. The mutant virus also showed higher intravenous pathogenicity in chickens. Based on the results, authors concluded that these mutations are crucial factors for the biological activities of genotype III NDV viruses. The context is simple and clear, however, following points should be revised/reconsidered.
Response: We appreciate the respected reviewer 1 for the useful comments. We have tried to consider all suggestions revised the manuscript based on the suggestions below.
Major comments
Point 1: Authors should strictly discuss the results based on the significant differences by statistical analyses at sections as follows.
L320-L323, Figure 2: HN with single mutations (A494D or E495K) showed no significant differences from their control, MukHN, in HAd, NA, and fusion activities. The description in L320-L323 should be revised.
L407-L410, Figure 6: Significant differences in F1/HN ratios were ONLY found in the case of MOI 1 viral infection. The description in L407-L410 should be revised.
L434, Figure 7B: Significant differences were found after 24 or 36 hpi, not 12 hpi.
Response 1: Thanks for the good suggestions, and we have revised them separately. (revision edition: L330-339, L426, and L451).
Ponit 2: L485-L487, L626-L628, L637: Tissue tropisms of the viruses were hard to discuss based on the present experimental infection study because the chickens were inoculated with the virus via direct intravenous route, not natural infection route. Authors should describe not ‘circulation route’ but the reason why the mutant viruses maintained in peripheral bloods longer than the original virus.
Response 2: Thanks for the good suggestion. We have revised the description based on the reason why the mutant viruses could maintain in peripheral blood longer than the original virus. (revision edition: L502-506, L650-656, and L659-664).
Point 3: The aa mutations at the 494 and/or 495 of HN proteins have been also found in other (genotype III) NDVs in the field? Such information/discussion should be added to make this article more significant.
Response 3: Thanks for the suggestion. Yes, we have disscussed the relevant researches about aa mutations at the 494 and 495 of NDV HN in L624-626. However, the research on the effects of aa mutations at 494 and/or 495 of NDV HN is relatively rare, and it is of great significance for the research in this field.
Minor comments
Ponit 4: Introduction: Additional explanation about NDV genotype is preferable because ‘Genotype III’ should be one of keywords of this paper.
Response 4: That’s a good suggestion. We have added the relevant explanation about NDV genotype in Introduction (revision edition: L54-L59) and ‘Genotype III’ in Keywords (revision edition: L47).
Point 5: L95, L106: ‘Kept in our lab’ and ‘provided by our lab’ are not appropriate description for academic article because readers are not able to reproduce your work. Please clarify backgrounds or previous reference works of these materials. For example, you isolated the viruses by yourself or obtained them from other institutes? Vectors were purchased/handmade/provided from others? etc.
Response 5: Thanks for the useful suggestions. We have revised the relevant contents in L101-102 and L113-114.
Point 6: Table 1: Recognition sites of restriction enzymes should be also shown in the table.
Response 6: Thanks for the good suggestion. We have shown the recognition sites of restriction enzymes in bold. (revision edition: Table 1 and L141-L142).
Point 7: Table 2: E494K -> E495K?
Response 7: Thank you for bringing this oversight to our attention. We agree with the reviewer and have made the proper correction accordingly in Table 2.
Point 8: L341: ‘*’ not be Italic.
Response 8: Thanks for the good suggestion. We have revised the relevant content in L357.
Point 9: L350-L355: Font is different from other sections.
Response 9: Thank you for bringing this oversight to our attention. We have made the proper correction accordingly in L366-L372.
Point 10: L426: ‘greatly’ should be deleted. Too subjective.
Response 10: As suggested by the reviewer, we have deleted ‘greatly’ in L443.
Point 11: Figure 8B: should be deleted. Same results as Figure 8A.
Response 11: Thank you. We have deleted Figure 8B accordingly.
Point 12: Figure 8C: not informative pictures. If you want to show these, detail explanations about symptoms should be added in the figure legends.
Response 12: That’s a valuable suggestion. We have added the detail explanations about clinical symptoms in the figure legends in L512-L514 and L516-L518.
Point 13: Discussion: should be divided into some paragraphs according to the contents.
Response 13: Thanks for the good suggestion. We have divided the discussion into five paragraphs according to the contents.
The above are all revisions of the paper, and thank you very much for your insightful comments and suggestions once again.
Reviewer 2 Report
Overall, this is a clear, concise, and well-written manuscript. The introduction is relevant and theory based. Sufficient information about the previous study findings is presented for readers to
follow the present study rationale and procedures. The methods, results and discussions are generally appropriate.
Author Response
Response to Reviewer 2 Comments
Comments to authors:
Overall, this is a clear, concise, and well-written manuscript. The introduction is relevant and theory based. Sufficient information about the previous study findings is presented for readers to follow the present study rationale and procedures. The methods, results and discussions are generally appropriate.
Response: We would like to thank the respected reviewer 2 for the good comments.